# The Odorant Binding Protein, SiOBP5, Mediates Alarm Pheromone Olfactory Recognition in the Red Imported Fire Ant, *Solenopsis invicta*

**DOI:** 10.3390/biom11111595

**Published:** 2021-10-28

**Authors:** Yuzhe Du, Jian Chen

**Affiliations:** 1Southern Insect Management Research Unit, Agricultural Research Service, United States Department of Agriculture, 141 Experiment Station Road, Stoneville, MS 38776, USA; 2Biological Control of Pests Research Unit, Agricultural Research Service, United States Department of Agriculture, 59 Lee Road, Stoneville, MS 38776, USA; jian.chen@usda.gov

**Keywords:** red imported fire ants, *Solenopsis invicta*, odor-binding proteins (OBPs), RNA interference (RNAi), electroantennography (EAG), qRT-PCR

## Abstract

Olfaction is crucial in mediating various behaviors of social insects such as red imported fire ants, *Solenopsis invicta* Buren. Olfactory receptor (OR) complexes consist of odor-specific ORs and OR co-receptors (Orco). Orcos are highly conserved across insect taxa and are widely co-expressed with ORs. Odorant binding proteins (OBPs) can transport semiochemicals to ORs as protein carriers and thus constitute the first molecular recognition step in insect olfaction. In this study, three OBP genes highly expressed in *S. invicta* antenna, OBP1, OBP5, OBP6, and Orco were partially silenced using RNA interference (RNAi). RNAi SiOBP5- and Orco-injected ants showed significantly lower EAG (electroantennography) responses to fire ant alarm pheromones and the alkaloid, 2,4,6-trimethylpyridine than water- or GFP-injected ants 72 h post injection. Subsequent qRT-PCR analysis demonstrated that the transcript level of the OBP1, OBP5, OBP6, and Orco significantly decreased 72 h after ants were injected with dsRNAs; however, there were no transcript level or EAG changes in ants fed dsRNAs. Our results suggest that *S. invicta* Orco and SiOBP5 are crucial to fire ants for their responses to alarm pheromones. RNAi knocking down SiOBP5 can significantly disrupt alarm pheromone communication, suggesting that disrupting SiOBP5 and Orcos could be potentially useful in the management of red imported fire ants.

## 1. Introduction

The red imported fire ant, *Solenopsis invicta* Buren, is a well-known global invasive ant species that was introduced into the United States from South America in the 1930s. It is currently found in many southern and western states and Puerto Rico [1]. Like other social insects, the maintenance of colony cohesiveness, sociality and defense depends on sophisticated pheromonal communication. Pheromones are crucial to the survival of ant colonies [2].

Insect olfaction is mediated by a subset of specialized cuticular structures called sensilla that are mainly found on the insect antennae and maxillary palp. Olfactory sensilla contain olfactory receptor neurons (ORNs), which are suspended in an aqueous lymph that harbors olfactory receptors (ORs) in their dendritic membranes. OR complexes consist of odor-specific ORs and OR co-receptors (Orco). Orcos are highly conserved across insect taxa, which are co-expressed with ORs. In addition, the sensillum lymph is characterized by water-soluble proteins called odorant binding proteins (OBPs) [3,4,5]. These polypeptides play an important role in the processes of chemo-detection and olfactory recognition, which can carry pheromones and odorants through the lymph’s hydrophilic barrier [3,4,5]. OBPs are small (they have a molecular weight of ~130–150 amino acids) ligand carrier proteins that include pheromone binding proteins (PBPs) and general odorant binding proteins (GOBPs) [6].

OBPs have been reported in a wide range of insect species, and the number of OBPs varies in different species, ranging from 12 in ants to at least 35 in Drosophila, 44 in silkworms, and more than 100 in some mosquitoes [7,8,9,10,11]. So far, the largest OBP families in the Hymenoptera were reported in the parasitoid wasp *Nasonia vitripennis,* in which 90 sequences that encode putative OBPs were identified [12]. In *S. invicta*, a 17-gene-member OBP family was identified in both antennal and non-antennal tissue that are specific to the expression of these OBPs [6]. Like other insects, fire ant OBPs are also involved in pheromone transportation, and these are expressed specifically in the antennae. The objective of this study was to identify and characterize pheromone-specific OBPs by selectively silencing key OBPs using RNA interference (RNAi) and evaluate the changes by using an electrophysiological approach. RNAi is a powerful molecular tool used for understanding various aspects of insect physiology, while electroantennography (EAG) is an effective tool used for assessing changes in olfactory responses to pheromones [13]. The pheromone 2-ethyl-3,6-dimethylpyrazine (EDP) is released by the mandibular gland of *S. invicta*, and is identified as the primary alarm pheromone [14]; however, which OBPs are involved in the transport of fire ant alarm pheromones has remained unknown. It was reported that SiOBP1, 5, and 6 were highly expressed in the antennae of *S. invicta* workers [6]. We attempted to knock down the SiOBP1, 5, and 6 genes—and thereby the Orco—using RNAi (through the injection or feeding of dsRNAs). The EAG response to alarm pheromones and several other EAG-active volatile compounds, and the transcript levels of the OBPs were then examined after the treatments.

## 2. Materials and Methods

### 2.1. Insects

Red imported fire ants were collected from Washington County, Mississippi. The colonies were maintained in Fluon-coated trays and kept in an insect rearing room at 26 °C. The social form of *S. invicta* colonies was determined using PCR on Gp-9 alleles [15]. All the ants used in this study were from monogyne colonies. The colonies were fed with 10% sucrose and frozen house crickets, *Acheta domesticus* L., and kept at room temperature with ~70% humidity and a 16:8 dark–light photoperiod.

### 2.2. Chemical

2-ethyl-3,5(6)-dimethyl pyrazine (EDP, a mixture of the 3,5- and 3,6-dimethyl isomers, herein referred to as pheromone isomers), E-β-Farnesene (EBF), 2,4,6-trimethylpyridine (TMP), and ylang ylang oil were purchased from Sigma-Aldrich (Sigma-Aldrich, St. Louis, MO, USA). The purities of these four chemicals were >95%.

### 2.3. Total RNA Extraction, cDNA and dsRNA Synthesis

Total RNA was isolated from the antennae of fire ant workers using the Trizol reagent RNA Isolation System (Invitrogen, Life Sciences, Carlsbad, CA, USA) following the manufacturer’s instructions. One microgram of *S. invicta* total RNA was used for cDNA synthesis with the SuperScript^TM^ II Reverse Transcriptase System (Invitrogen, Life Sciences, Carlsbad, CA, USA). One microliter of the cDNA sample was used as a template, and the SiOBP1, SiOBP5, SiOBP6, and SiOrco fragments were amplified using RT-PCR with specific primers (Table 1) and conjugated using a 20 bp T7 RNA polymerase promoter. One microgram of each PCR product (SiOBP1, SiOBP5, SiOBP6, and SiOrco—Figure 1A) was used as a template for dsRNA synthesis with the MEGAscript T7 System (Ambion, Austin, TX, USA). The dsRNA was phenol: chloroform, which was extracted, isopropanol-precipitated, re-suspended in RNase-free water, and quantitated at 260 nm using a NanoDrop 2000 Spectrophotometer (Thermo Fisher Scientific Inc., Carlsbad, CA, USA). Green fluorescence protein (GFP) dsRNA served as the negative control and was synthesized using a 546 bp GFP DNA template that was amplified by the primers also shown in Table 1 using the same kit as above. The quality and integrity of the dsRNAs were determined using agarose gel electrophoresis (Figure 1B).

### 2.4. dsRNA Injection and Feeding

Different time point experiments were conducted by using dsRNA feeding or injection on the major fire ant worker, with each using the same experimental design and treatments. At 48, 72 and 96 h post injection and feeding, treated worker ants were collected for EAG tests, and, once EAG bioassays were finished, the antennae were immediately immersed in RNALater (Invitrogen, Thermo Fisher Scientific, Waltham, MA, USA), and stored at −80 °C in a freezer until RNA extraction. For the dsRNA feeding experiment, treatment replication was achieved over three independent experiments. Fifty microliters each of the OBP dsRNA, Orco dsRNA, and GFP dsRNA were diluted about 20-fold in sucrose water in a 1.5 mL Eppendorf tube, and the final concentration was 0.2 μg/μL. A cotton ball was used to plug the tube and also provided the ants with a feeding platform. For the dsRNA injection experiments, there were four treatments (the dsRNA of OBP1, OBP5, OBPB6, and Orco) and three controls (non-, water-, and GFP-injection), and each treatment had three experimental replicates. The major adult workers were placed in a petri dish (10 cm, containing a layer of 1% agarose), which was placed in a bucket of ice to immobilize the ants for injection. The OBP dsRNA was injected into the hemocoel of *S. invicta* adult workers using a Nanoinjector II System (Drummond Scientific, Broomall, PA, USA) with a pulled borosilicate needle (World Precision Instruments, Sarasota, FL, USA). The connective membrane between the third and fourth dorsal abdominal segments of each ant was pierced for the delivery of 50 nL of dsRNA (200 ng) in each treatment. The OBP dsRNA (200 ng or 50 nL/ant), Orco dsRNA (200 ng or 50 nL/ant), GFP dsRNA (200 ng or 50 nL/ant) and water (50 nL/ant) were each injected into 50 *S. invicta* major workers. Five live ants per replicate were tested for EAG, and thus 10 antennae were collected and pooled at each time point post injection or feeding.

### 2.5. Electroantennography (EAG) Assays

The procedures for electroantennography (EAG) and odorant delivery were identical to those previously described [16,17,18,19]. The sham-(control) and RNAi-treated ants were randomly chosen for the EAG test. 2-ethyl-3,5(6)-dimethyl pyrazine (EDP), E-β-Farnesene (EBF), 2,4,6-trimethylpyridine (TMP), and ylang ylang oil were diluted with pentane; only one concentration (10% *v*/*v*) for each odorant stimulus was used to evaluate the EAG changes in the RNAi-treated ants.

### 2.6. Real-Time Quantitative RT-PCR

We performed real-time quantitative reverse transcriptase PCR (qRT-PCR) to determine the mRNA expression level of SiOBP1, SiOBP5, SiOBP6, and SiOrco. GAPDH, an internal reference transcript gene of *S. invicta*, was used for initial normalization and was quantified. The real-time quantitative RT-PCR procedures used were identical to those of Zhang et al. [6]. Total RNA was extracted from each pool of ant antennae using Trizol and the RNA quantity was analyzed using a NanoDrop 2000 Spectrophotometer (Thermo Fisher Scientific Inc., Carlsbad, CA, USA) at 260 nm. The first cDNA strand of each pool was synthesized from 1 µg of total RNA using the SuperScript^TM^ II Reverse Transcriptase System (Invitrogen, Life Sciences). Gene-specific primer pairs were designed using Primer 3 (Table 1), and 9 µL of cDNA (diluted 1:50) was used as a template in a 20 µL SYBR Green Supermix reaction (Bio-Rad Inc. Hercules, CA, USA). Quantitative PCR was performed in a two-step amplification with 40 cycles of 95 °C for 30 s and 56 °C for 30 s using a Bio-Rad IQ thermocycler (Bio-Rad Laboratories Inc. Hercules, CA, USA). The normalized abundance of the target RNA (OBP and Orco) compared to the internal reference RNA (GAPDH) was calculated (ΔCT) for the treatment and the control (i.e., non-injection or sucrose-water-feeding) samples. The relative abundance of the target RNA (OBP and Orco) in the treatment (dsRNA-OBPs and Orco) compared to the control (non-injection or sucrose-water-feeding) was calculated using the 2-ΔΔCT relative expression method. For this calculation, the normalized target RNA value for each treatment sample was divided by the average normalized values for the control samples. For a comparative illustration, the relative expression of GAPDH in each of the control samples was calculated in the same manner.

### 2.7. Statistical Analysis

The statistical significance of the differences between the RNAi-treated groups for each of qRT-PCR and EAG was determined by using a one-way analysis of variance (ANOVA)—with the least significant difference (LSD) test for qRT-PCR and Tukey’s HSD test for EAG analysis—and significant values were set at *p* < 0.05. All statistical analyses were performed using SPSS (version 26.0; SPSS Inc., Chicago, IL, USA).

## 3. Results

We aimed to identify which OBP(s) are involved in detecting and transporting the alarm pheromones of *S. invicta*. Among 17 *S. invicta*, the OBPs SiOBP1, SiOBP5, and SiOBP6 exhibited antenna-enriched expression [6]; we initially hypothesized that these three OBPs might be involved in pheromone detection in *S. invicta*. Thus, RNAi-treated ants were fed or injected with the dsRNAs SiOBP1, SiOBP5, SiOBP6, SiOrco, GFP and with water.

We then examined the EAG responses of sham- and RNAi-treated ants to four odorant stimuli: 2-ethyl-3,5(6)-dimethyl pyrazine (EDP), E-β-Farnesene (EBF), 2,4,6-trimethylpyridine (TMP), and ylang ylang oil. To validate the altered EAG responses of the SiOBP1, SiOBP5, SiOBP6, and Orco dsRNA-injected major fire ant workers, EAG responses from five ants of each experimental group (i.e., non-injection, water-injection, dsGFP-injection, Orco-injection, and three SiOBP RNAi-injection) were recorded at 48, 72, and 96 h post injection. The odorant stimuli EDP, TMP, and ylang ylang oil elicited significant EAG responses in non-injected, water-injected, and dsGFP-injected ants, but EBF evoked no significant EAG response (Figure 2A). Furthermore, the SiOBP5 and Orco RNAi-injected ants showed significantly reduced responses to EDP, TMP, and ylang ylang oil compared with non-injected, water-injected, and dsGFP-injected control ants at 72 and 96 h post injection (Figure 2B,C). Because injecting dsRNA severely damaged these ants, some ants were less mobile at 96 h post injection than normal ants. Therefore, only the EAG responses at 72 h post injection are shown (Figure 2). In contrast, four dsRNA-fed ants exhibited no significant EAG changes compared with those of the sucrose-water- and dsGFP-fed ants at any time point (Figure 3). Compared with the dsRNA-injected ants, the dsRNA-fed ants were very active, even 120 h post feeding. We also noted that the SiOBP1- and SiOBP6-injected ants exhibited no significant EAG changes to EDP, TMP, or ylang ylang oil. Nevertheless, EBF did not elicit any significant responses from the sham and RNAi-treated ants (Figure 3).

Once the EAG bioassays were finished, we applied a combination of RT-PCR and real-time quantitative PCR (qPCR) to determine the mRNA levels of SiOBP1, SiOBP5, SiOBP6, and Orco in the antennae of RNAi-treated (dsSiOBPs-injected and dsSiOrco-injected) and control (non-injected, water-injected and dsGFP-injected) ants using GAPDH as an internal reference transcript gene. The SiOBP dsRNA-injected ants displayed a reduction in SiOBP1, SiOBP5, SiOBP6, and Orco transcript levels (50.2%, 42.8%, 46.6%, and 55.1%, respectively) when compared with the water-injected ants (average 96.3%), dsGFP-injected (average 95.6%) ants, and non-injected controls (normalized to 100%) (Figure 4A). The RT-PCR analysis showed a significant reduction in SiOBP1, SiOBP5, SiOBP6, and Orco transcript levels in the dsRNA-injected ants, as compared with the water-injected or GFP-injected ants. Furthermore, the water-injected or GFP-injected and non-injected ants displayed almost same levels of transcripts, indicating that RNAi dsRNA injection is responsible for the reduction in the SiOBP1, SiOBP5, SiOBP6, and Orco mRNA transcript levels. This partial silencing of SiOBP1, SiOBP5, SiOBP6, and Orco shown by qPCR analysis also demonstrates the feasibility of using the RNAi approach to significantly reduce the three antenna-enriched expression of OBPs and Orco genes. By contrast, the SiOBP and Orco dsRNA-fed ants displayed no significant transcript level changes when compared with both the sucrose-water-fed (normalized to 100%) and dsGFP-fed (average 96.8%) ants (Figure 4B). The correlation with EAG data (Figure 2) also suggests that the ~50% transcript reduction in RNAi dsRNA-injected ants is sufficient to generate reduced EAG responses to EDP, TMP, and ylang ylang oil; correspondingly, no transcript level changes in RNAi-fed ants elicited any EAG responses to the four odorant stimuli (Figure 3).

## 4. Discussion

In this study, we demonstrated that SiOBP5 and Orco dsRNA injection resulted in a significant reduction in the EAG responses of *S. invicta* to an alarm pheromone EDP; an alkaloid, TMP; and an essential oil, ylang ylang oil. The results indicated that SiOBP5 is the binding protein involved in the capture and transport of EDP, TMP, and the active compound(s) in ylang ylang oil to the receptors in the antennae of *S. invicta*. Although only a 42–55% silencing of SiOBP1, SiOBP5, SiOBP6, and Orco was achieved 72 h post injection, the partial silencing clearly affected the antennal EAG responses to EDP, TMP, and ylang ylang oil, which suggested that the fire ant olfactory system is remarkably sensitive [20]. That may be the reason why the partial transcript reduction achieved in our experiments was significant enough to affect EAG responses to EDP, TMP, and ylang ylang oil. By contrast, the reduced SiOBP1 and SiOBP6 transcripts did not affect the EAG responses to EDP, TMP, and ylang ylang oil. These two OBPs may be involved in the detection and transport of other chemical compounds. Nevertheless, there were no significant differences in the electrophysiological response to the aphid’s alarm pheromone EBF between the sham and RNAi treated ants, indicating that the three tested OBPs may not be involved in the olfactory detection of this compound by *S. invicta.*

The down regulation of the *S. invicta* Orco’s and three highly expressed antenna OBPs’ (SiOBP1, SiOBP5, and SiOBP6) transcript levels using RNAi injection severely affected the *S. invicta* olfactory signal transduction and made some ants less mobile than normal, but the RNAi-fed ants were very active even 120 h post feeding. Unfortunately, the ants fed with dsRNA SiOBPs, and Orco did not show any effects on the transcription and EAG levels at all. Feeding with dsRNA was reported to be a successful method for RNAi in some sap-sucking insects, such as the pea aphid (*A. pisum*) [21,22], white fly, (*Bemisia tabaci*) [23], and triatomine bug (*Rhodnius prolixus*) [24]. Even in *S. invicta*, the effect of the regulation of the pheromone-biosynthesis-activating neuropeptide (PBAN) on *S. invicta* survival has been investigated using RNAi. The PBAN dsRNA interfered with the melanization (cuticle darkening) of pupae after injection and larvae mortality increased when dsRNA was fed in a sucrose solution to nurse ants [25]. Additionally, these dsRNA-PBAN-injected ants showed decreased transcription and the inability to follow trail pheromones [26].

Our results indicate that the RNAi Orco dsRNA-injection resulted in a significant reduction in the transcript level and electrophysiological recording of the response to EDP, TMP, and ylang ylang oil. Insect Orcos are highly conserved, not only in *S. invicta*, and the RNAi silencing the olfactory Orco also reduced the response to pheromones in other insects, including the red palm weevil (*Rhynchophours ferrugineus*) [27], *Grapholita molesta* [28], *Protaetia brevitarsis* Lewis [29] *Blattella germanica* [30], aphids [31], subterranean termite [32], *gypsy moth* [33], *Rhodnius prolixus* [34], *Bactrocera dorsalis* [35], and *Apolygus lucorum* [36]. Orco silencing may provide a novel approach for managing various pest insects, including pest ants.

Unlike SiOBP5, silencing SiOBP1 and SiOBP6 did not result in any EAG changes in response to EDP, TMP, and ylang ylang oil. These three SiOBPs were highly expressed in *S. invicta* antennae. In a total of 17 SiOBPs, SiOBPs 1, 5, 6, 9, 10, and 11 have various orthologous OBPs in different insects, especially SiOBP5 and SiOBP6 [6]. A limited phylogenetic tree of OBP protein sequences that includes the genomes of two ant species (the leaf cutter ant, *Acromrymex echinatior*, and Indian/Jordon jumping ant, *Harpegnathos saltator*) and the European honeybee (*Apis mellifera*) illustrated that SiOBP5 is highly conserved with AeOBP6, HsOBP5X1, HsOBP5X2, and AmOBP6, and that SiOBP6 is conserved with AeOBP5, HsOBP7, and AmOBP6/8. Since EDP, TMPs and ylang ylang oil also elicited strong EAG responses in *A. mellifera* [17], we therefore speculate that the SiOBP5 orthologue AmOBP5 may be the binding protein that aids in the transportation of EDP, TMP, and ylang ylang oil in the *A. Mellifera*. Although EBF could not elicit significant EAG responses in either the SiOBP RNAi-treated or -untreated ants, EBF evoked robust olfactory responses in *S. invicta* workers and female alates-specific basiconica sensilla in single-sensillum recording (unpublished). Which SiOBPs bind to EBF in *S. invicta*? The answer is very likely the orthologue of ApisOBP7. In aphid species, ApisOBP7 is the OBP responsible for mediating the transportation of the aphid alarm pheromone EBF [32,37]. Further analyses need to be performed on the structure and ligand-binding capability of the identified candidate OBPs.

RNAi in combination with EAG approaches are widely used and well-accepted methods for the characterization of OBPs [38,39,40,41]. Such attempts have confirmed the role of OBPs in olfaction as carriers of pheromones through the aqueous environment of the sensillum lymph to ORs [3]. Our study demonstrates that EDP, TMP, and ylang ylang oil transportation disruption can occur through SiOBP5 silencing. EAG recordings indicated that SiOBP5 silencing in fire ants decreases their response to EDP, TMP, and ylang ylang oil. We observed a perfect correlation between the reduction in SiOBP5 transcript levels and the reduction in antennal responses to the alarm pheromone, which is straightforward evidence that SiOBP5 is involved in the detection of alarm pheromones by fire ants.

Currently, synthetic insecticides are commonly used in controlling fire ants. However, synthetic insecticides do not always deliver satisfactory control. The RNAi technique aimed at disrupting the olfactory system has the potential to interfere with such critical behaviors as foraging and mate location, ultimately disturbing the reproductive process and decreasing the *S. invicta* population. The RNAi technique has shown considerable potential for controlling insect pests [13]. To enable the use of the RNAi technique in managing *S. invicta*, having a feasible dsRNA delivery method is a must. Delivery through feeding is desirable because it is easy to incorporate into the current bait technology. A variety of new technologies might offer excellent future prospects for using RNAi for managing ants, such as nanoparticles, engineered microorganisms, transgenic bacteria that express dsRNA, the mass production of siRNA, and advanced formulations [13].

## Figures and Tables

**Figure 1 biomolecules-11-01595-f001:**
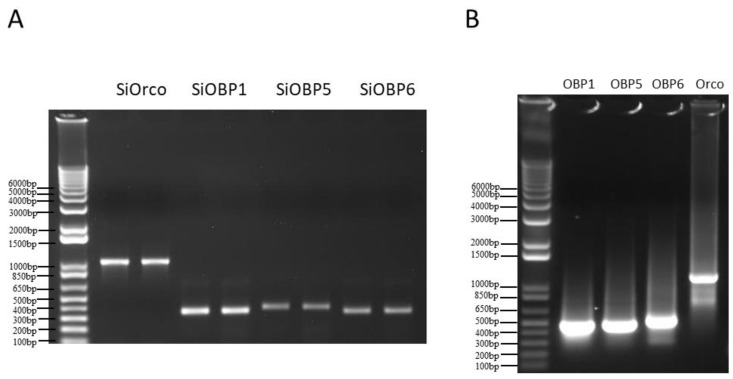
(**A**) The electrophoresis gel of each PCR product (SiOrco, SiOBP1, SiOBP5, and SiOBP6), which was used as a template for the dsRNA synthesis. (**B**) The electrophoresis gel of each dsRNA of SiOBP1, SiOBP5, SiOBP6, and SiOrco, which was used for feeding and injection.

**Figure 2 biomolecules-11-01595-f002:**
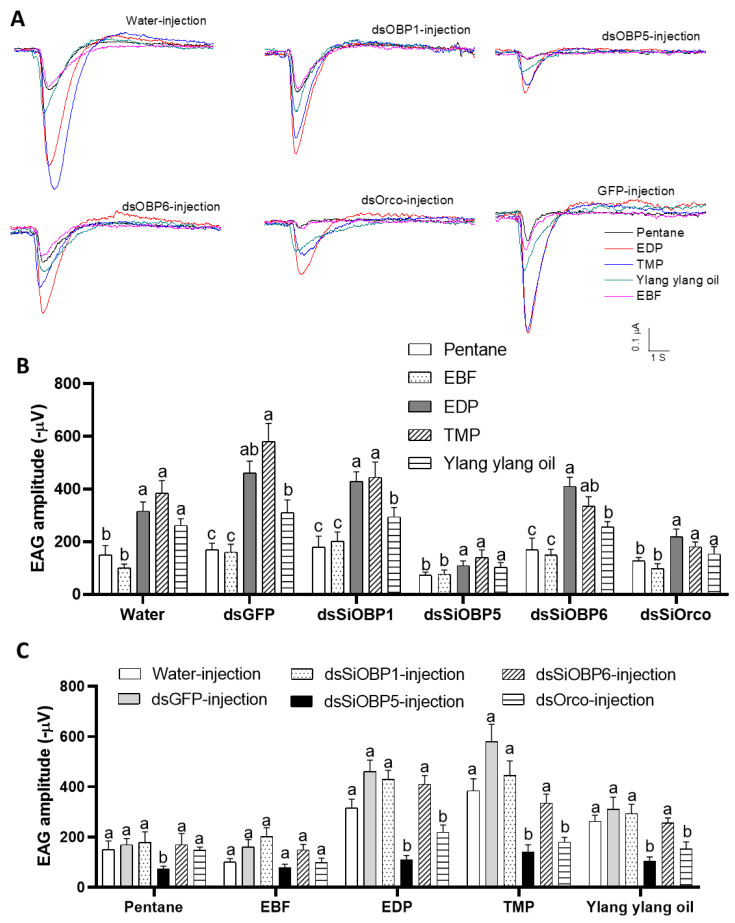
The EAG responses of dsSiOBP1-, dsSiOBP5-, and dsSiOBP6- and dsSiOrco-injected fire ant workers. (**A**) The representative EAG traces elicited by EDP, EBF, TMP, and ylang ylang oil 72 h post injection with water, dsGFP, dsSiOBP1, dsSiOBP5, dsSiOBP6, and dsSiOrco. (**B**) The EAG responses of water, dsGFP, dsSiOBP1, dsSiOBP5, dsSiOBP6, and dsSiOrco 72 h post injection. (**C**) The EAG responses to pentane, EBF, EDP, TMP, and ylang ylang oil 72 h post injection with the dsRNAs. The different letters on the top of the bars indicate statistically significant differences, as determined by one-way analysis of variance with Tukey’s HSD test, and significance was set at *p* < 0.05. (**B**,**C**) reflect the same data.

**Figure 3 biomolecules-11-01595-f003:**
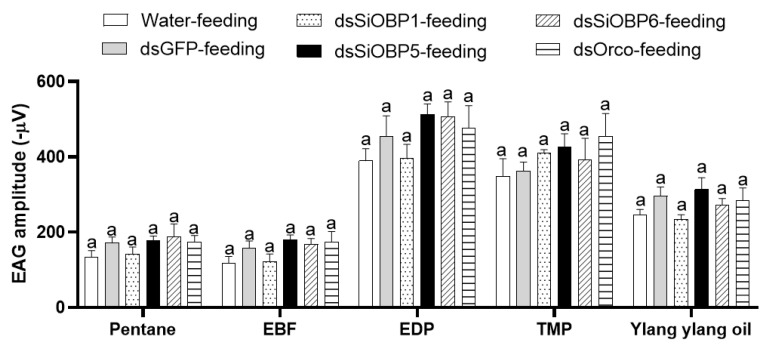
The EAG responses of fire ant workers to pentane, EBF, EDP, TMP, and ylang ylang oil 72 h post feeding with sucrose water, dsGFP, dsSiOBP1, dsSiOBP5, dsSiOBP6, and dsSiOrco. The same letters on the top of the bars indicate statistically significant differences, as determined by one-way analysis of variance with Tukey’s HSD test with significance set at *p* < 0.05.

**Figure 4 biomolecules-11-01595-f004:**
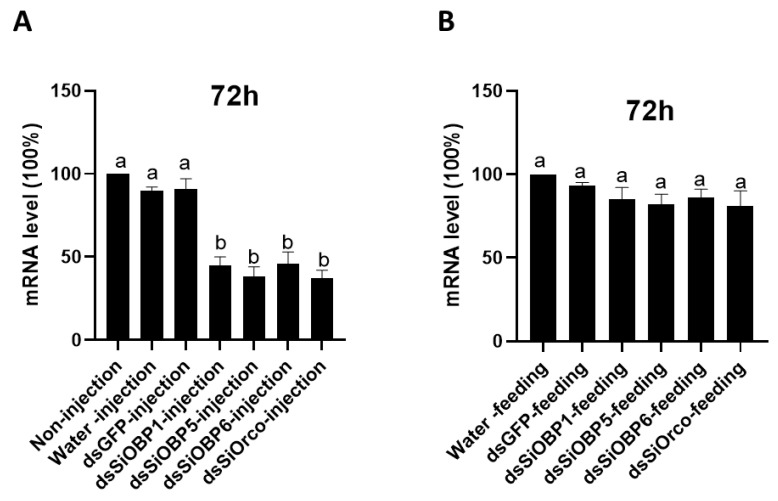
RT-PCR analysis showed a clear reduction in SiOBP1, SiOBP5, SiOBP6, and Orco transcript levels in dsRNA-injected ants. (**A**)—mRNA expression levels in non-injected, water-, dsGFP-, dsSiOBP1-, dsSiOBP5-, dsSiOBP6- and dsSiOrco-injected ants 72 h post injection. (**B**)—mRNA expression levels in sucrose-water-, dsGFP-, dsSiOBP1-, dsSiOBP5-, dsSiOBP6- and dsSiOrco-fed ants 72 h post feeding. The different letters on the top of the bars indicate statistically significant differences, as determined by one-way analysis of variance with the least significant difference (LSD) test, with significance set at *p* < 0.05.

**Table 1 biomolecules-11-01595-t001:** A list of the primer sequences used for the dsRNA synthesis and real-time quantitative reverse transcriptase PCR (qRT-PCR).

Gene	Primer Use	Sequence (5′–3′)
GFP	dsRNA synthesis	T7F: TAATACGACTCACTATAGGG CCATCCTGGTCGAGCTGGACGGCG
		T7R:TAATACGACTCACTATAGGG TCCAGCAGGACCATGTGATCGCGC
OBP1	dsRNA synthesis	T7F: TAATACGACTCACTATAGGG AGTGTTGCTACAGGCGATTTA
		T7R: TAATACGACTCACTATAGGG CTTGTACGGCGCATTTGTTC
OBP5	dsRNA synthesis	T7F: TAATACGACTCACTATAGGG ATGTGCGCGCGTTTACTA
		T7R: TAATACGACTCACTATAGGG AAAGGACCCGCCATTTCA
OBP6	dsRNA synthesis	T7F: TAATACGACTCACTATAGGG TCGCTTGTGTTTATGGCAAATC
		T7R: TAATACGACTCACTATAGGG TATGCTAACGCACACCCTTC
OrCo	dsRNA synthesis	T7F: TAATACGACTCACTATAGGG GTTTCGCGCTTCTACTCCAC
		T7R: TAATACGACTCACTATAGGG CTCCGATCCATCGTACCAGT
OrCo	qRT-PCR	F: GGTCCGATCGTTCTATCCATTC R: CCAACGAATTGGCGTTGATAAG
GAPDH	qRT-PCR	F: AAGCTGTGGCGTGATGGCG R: AGGAGGCAGGCTTGGCGAG
GFP	qRT-PCR	F: AAGCTGTGGCGTGATGGCCG R: AGGAGGCAGGCTTGGCGAGT
OBP1	qRT-PCR	F: CGATGGGAAACTCTCGAATGA R: CAGCATCTCAGCTTCCAAATTAC
OBP5	qRT-PCR	F: GGACGCATAGATGACGGTATG R: TGAGCAATCATCGCCAGTAG
OBP6	qRT-PCR	F: CGCTTGTGTTTATGGCAAATCT R: TTTGATCCTTGCCCACTCTATC

## Data Availability

Not applicable.

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
