# Peer review of "The Odorant Binding Protein, SiOBP5, Mediates Alarm Pheromone Olfactory Recognition in the Red Imported Fire Ant, Solenopsis invicta"

_biomolecules, 2021, doi:10.3390/biom11111595_

Round 1

Reviewer 1 Report

In this work, Yuzhe Du and Jian Chen investigated the roles of three OBPs and one Orco in the reception of alarm pheromone in red imported fire ants, Solenopsis invicta Buren. RNAi of dsRNA injection suggested that OBPs and Orco were significantly reduced at the transcriptional levels. However, only knocking down OBP5 and Orco changed EAG responses, suggesting that these two genes play key roles in recognizing alarm pheromone. In general, this work is well designed, the results are clear, and the discussion is appropriate. I just have few minor concerns as follows.

  1. The authors knocked down one Orco. Is this Orco unique in the fire ant? How many Orcos do exist in this ant?
  2. Line 85, If the authors have the data, please show it. Currently, more and more journals no longer accept the expression of “data not shown”. I really agree with that.
  3. Lines 146-149, Can the authors provide any data supporting the conclusion on mobile behavior changes?
  4. Line 225, I think the authors should mention what fold they diluted the cDNA.
  5. Figure S1, the authors should label the molecular weight of the DNA ladder.
  6. Line 276, perhaps the authors collected the antennae for RNA extraction and qPCR. I think it should be clarified.
  7. Line 286, Yes, GAPDH is generally used as an internal reference transcript gene, but it is not always stable in some tissues and conditions. Have the authors tested if this gene is suitable for their experimental conditions? Additionally, the normalization method of qPCR data should be mentioned.

Author Response

Review 1

  1. The authors knocked down one Orco. Is this Orco unique in the fire ant? How many Orcos do exist in this ant?

Response: Yes, the Orco is indeed unique for fire ant, although Orcos are highly conserved across insect taxa. Each insect species has only one Orco. ORs complexes consist of odor specific ORs and Orco.

  1. Line 85, If the authors have the data, please show it. Currently, more and more journals no longer accept the expression of “data not shown” I really agree with that.

Response: We have the data, which was shown in Figure 2.

  1. Line 146-149, Can the authors provide any data supporting the conclusion on mobile behavior changes?

Response: Some ants were less mobile than normal ants, most likely due to the damage from RNAi injection. The connective membrane between the third and fourth dorsal abdominal segments was pierced when injecting 50 nL of dsRNA into each ant.

  1. Line 225, I think the authors should mention what fold they diluted the cDNA.

Response: The concentration of each dsRNA was about 4000ng/μl, which was diluted about 20-fold in sucrose water in a 1.5-ml Eppendorf tube, and the final concentration was 0.2 mg/ml.

  1. Figure S1, the authors should label the molecular weight of the DNA ladder.

Response: The molecular weight of the DNA ladder was added.

  1. Line 276, perhaps the authors collected the antennae for RNA extraction and qPCR. I think it should be clarified.

Response: The methods of antennae collection for RNA extraction and qPCR have been added

  1. Line 286, Yes, GAPDH is generally used as an internal reference transcript gene, but it is not always stable in some tissues and condition. Have the authors tested if this gene is suitable for their experimental conditions? Additionally, the normalization method of qPCR data should be mentioned.

Response: GAPDH is stable for our experimental condition. The normalization method of qPCR data has been added.

Reviewer 2 Report

This paper by Du and Chen used RNAi to knockdown 3 OBPs and the OR co-receptor Orco in the fire ant, Solenopsis invicta and showed that one of these OBP, SiOBP5, has reduced response to fire ant alarm pheromones. Orco, which is needed for all olfactory functions in insects, showed similar response when knocked down by RNAi. In addition, the authors used two RNAi methods and showed that injecting fire ants with dsRNA can reduce transcript levels but feeding these ants with dsRNA does not, suggesting that moving forward, injection with dsRNA is the way to manipulate RNA levels for functional studies in this insect. This is an important study where the authors identified one OBP which is involved in alarm pheromone response. This is likely to be important in the control of this invasive species. I have some minor suggestions

1) The title should be changed from “The fire ant odorant binding protein mediates alarm pheromone olfactory recognition” to “The odorant binding protein, SiOBP5,  mediates alarm pheromone olfactory recognition in the red imported fire ant, Solenopsis invicta Buren” to show the exact gene that they discovered in mediating alarm pheromone response in this species. The original title is too broad and does not give any information.

2) Line 168, change “not like” to “unlike”

3) Figure 1c: Pentane and EBF means do not have letters from the statistical analysis.

Author Response

  1. The title should be changed from “ The fire ant odorant binding protein mediates alarm pheromone olfactory recognition” to “The odorant binding protein, SiOBP5, mediates alarm pheromone olfactory recognition in the red imported fire ant, Solenopsis invicta Buren” to show the exact gene that they discovered in mediating alarm pheromone olfactory recognition in the red imported fire ant, Solenopsis invicta Buren” to show the exact gene that they discovered in mediating alarm pheromone response in this species. The original title is too broad and does not give any information.

Response: We changed the title to “The odorant binding protein, SiOBP5, mediates alarm pheromone olfactory recognition in the red imported fire ant, Solenopsis invicta

  1. Line 168, change “not like” to “unlike”

Response: Corrected

  1. Figure 1c: Pentane and EBF means do not have letters from the statistical

Response: Added
